# Suicidal Ideation, Planning, and Attempts among Canadian Coast Guard and Conservation and Protection Officers

**DOI:** 10.3390/ijerph192316318

**Published:** 2022-12-06

**Authors:** Jolan Nisbet, Laleh Jamshidi, Katie L. Andrews, Taylor A. Teckchandani, Jill A. B. Price, Rosemary Ricciardelli, Gregory S. Anderson, R. Nicholas Carleton

**Affiliations:** 1Canadian Institute for Public Safety Research and Treatment (CIPSRT), University of Regina, Regina, SK S4S 0A2, Canada; 2Fisheries and Marine Institute, Memorial University of Newfoundland, Saint John’s, NL A1C 5R3, Canada; 3Dean Faculty of Science, Thompson Rivers University, Kamloops, BC V2C 0C8, Canada

**Keywords:** suicidal thoughts and behaviors, public safety personnel (PSP), mental health disorders, Posttraumatic Stress Injury (PTSI)

## Abstract

The current study provides estimates of suicidal ideation, planning, and attempts among Canadian Coast Guard personnel and Canadian Conservation and Protection Officers. Participants (*n* = 385; 59% men) completed a self-report survey that collected past-year and lifetime estimates of suicidal ideation, planning, attempts, sociodemographic information, and symptoms related to mental health disorders. Within the sample, participants reported lifetime suicidal ideation (25.7%), planning (10.9%), and attempts (5.5%). Participants reported past-year suicidal ideation (7.5%), planning (2.1%), and the percentage of attempts was too marginal to report due to ethical considerations. Canadian Coast Guard personnel and Conservation and Protection Officers reported higher percentages of past-year and lifetime suicidal ideation, planning, and attempts than the Canadian general population, but the percentages reported are comparable to other Canadian PSP sectors. No statistically significant results were observed for the sociodemographic covariates within the past-year, whereas, statistically significant results were observed for the lifetime sociodemographic covariates of age, province of residence, and previous work experience. There were statistically significant associations between past-year suicidal ideation and positive screens for major depressive disorder (MDD) and general anxiety disorder (GAD); as well, past-year suicidal planning was associated with social anxiety disorder (SAD). There were also statistically significant associations between lifetime suicidal ideation, planning, and attempts and positive screens for posttraumatic stress disorder (PTSD), MDD, and SAD. Lifetime suicidal ideation and planning were associated with panic disorder (PD). The current estimates provide valuable information for clinicians and stakeholders involved in prevention programs, treatment, or future research.

## 1. Introduction

Suicide is a complex human behaviour which tends to involve multiple causes [1]. There are diverse factors associated with death by suicide, including suicidal ideation (i.e., thoughts that may or may not include a plan to die), and suicidal behaviours (i.e., suicidal planning, and attempts) [2,3,4]. Risk factors for suicidal ideation, planning, and attempts may include screening positive for a mental health disorder, abuse suffered as a child, various interpersonal conflicts, insomnia, hopelessness, or substance abuse [4,5,6]. People experiencing suicidal ideation may not seek formal care due to wishing to handle the issue alone, facing structural constraints, or fearing stigma [7].

Stigma related to mental health concerns may be higher within certain professions, such as those of public safety personnel (PSP) [8,9]. The term PSP encompasses a diverse group of professionals who work to ensure the safety and security of Canadians, including, but not limited to, border services officers, correctional workers, firefighters (career and volunteer), Indigenous emergency management, operational and intelligence personnel, paramedics, policing (municipal and provincial), public safety communication, and Royal Canadian Mounted Police [10,11]. PSP also include professional personnel with duty-specific responsibilities related to law enforcement, environmental protection, or search and rescue operations. For example, the Canadian Department of Fisheries and Oceans (DFO) employs Conservation and Protection (C&P) Officers who have PSP duty-specific responsibilities related to law enforcement and habitat protection within Canadian waters. The DFO employs more than 600 C&P Officers who operate across seven regions in Canada. C&P Officers have specific training to fulfill a wide range of duties, either overtly or covertly, on land or at sea. C&P Officers may need to operate in remote locations with marginal to no backup. Canadian Coast Guard (CCG) personnel also fulfill PSP duty-specific responsibilities that involve search and rescue operations, monitoring for nautical hazards, responding to marine pollution, and providing navigational warnings within four Canadian regions to ensure sovereignty and security in Canadian waters [12,13,14].

C&P Officers and the CCG personnel report experiencing a high number of organizational stressors (e.g., bureaucratic red tape, staff shortages, excessive administrative duties), operational stressors (e.g., finding time to stay in good physical condition, fatigue, paperwork), and frequent exposure to potentially psychologically traumatic events (PPTE) [15,16]. A PPTE is a stressful event that may cause an individual psychological trauma [10]. The most frequent types of PPTE exposures include a serious transportation accident; a serious accident at work, home, or during recreational activity; physical assault; or sudden death, with most respondents reporting more than ten exposures to various PPTEs [16]. High levels of organizational and operational stressors, and frequent exposures to PPTE, place individuals at an elevated risk of developing symptoms related to one or more mental health disorders, which may increase an individual’s risk of developing suicidal behaviours [17,18,19,20].

Despite indications of increased risk for suicidal ideation, planning, and attempts, as in other PSP sectors, the literature focused on CCG personnel and C&P Officers is limited, with no published results regarding suicidal behaviours [19]. Comparisons may be possible with coast guard personnel from the United States Coast Guard (USCG) due to similar occupational duties; however, there may be important differences because the USCG is overseen by the Department of Defence, whereas in Canada, CCG personnel operate as a special agency within Fisheries and Oceans Canada [14]. In 2018, the USCG estimated the past-year suicidal ideation (4.7%), planning (1.6%), and attempts (0.2%) [21]. Previous research emphasised that high levels of secrecy within the USCG and ongoing mental health stigma may prevent personnel from seeking help, alongside adverse interpersonal or professional costs [22].

The current study was designed to address extant gaps in the literature regarding suicidal behaviours among C&P Officers and CCG personnel by: (1) assessing self-reported past-year and lifetime percentages of suicidal ideation, planning, and attempts among a diverse sample of CCG personnel and C&P Officers; (2) comparing suicidal ideation, planning, and attempts across sociodemographic categories; (3) assessing for associations between positive screens for mental health disorders (i.e., posttraumatic stress disorder (PTSD), major depressive disorder (MDD), generalised anxiety disorder (GAD), social anxiety disorder (SAD), panic disorder (PD), alcohol use disorder (AUD)) and suicidal ideation, planning, and attempts in the past year or lifetime [19,23,24]. Based on suicide-specific research within the general population and other PSP sectors, we expected CCG personnel and C&P Officers to report higher levels of suicidal ideation, planning, and attempts than the general population [19,25,26,27]. Women CCG personnel and C&P Officers were expected to be more likely to screen positive for mental disorders and more likely to report suicidal behaviours [19,27,28]. The study results can inform and support ongoing proactive efforts, as well as treatment, intervention, and research initiatives specifically for C&P Officers and the CCG personnel.

## 2. Materials and Methods

### 2.1. Procedure

The current study used data collected via a web-based self-report survey that was available in English or French. The study was approved by the University of Regina Institutional Research Ethics Board (REB# 2021-003; approved on 2 May 2021). All participants were provided with a plain language summary of the research and completed informed consent. The self-report survey was based on a set of validated measures used in a previous study of PSP [29] but collaboratively redesigned by the research team and the DFO/CCG team to ensure variables relevant to the DFO/CCG were included. The survey was promoted and distributed by the CCG/DFO via member unions through emails, social media posts, and a video encouraging participation. The self-report survey was available from 1 February 2021 to 31 January 2022. Participants completed the survey anonymously and were provided with a randomly generated unique code upon entering the survey to facilitate repeated entry into the survey and multiple survey sessions to accommodate full participation.

### 2.2. Data and Sample

A total of (*n* = 385) C&P Officers (27.8%) and CCG (71.7%) completed the self-report survey questions related to suicide; more specifically, the participants answered “yes” or “no” to past-year and lifetime suicidal ideation, planning, and attempts. Participants who completed at least 30% or more of the survey were included in the current analyses and results. Participants were mostly male (59.7%), identifying as men (59.0%). Most participants reported being 30 to 39 or 40 to 49 years old (28.8%, 28.1%), Caucasian (88.1%), married or in common-law relationships (68.1%), from British Colombia (56.4%), and with post-secondary education (e.g., Trade School; 2-Year College Diploma) (39.7%). Most participants did not have any previous work experience within the Canadian Armed Forces (CAF) or another PSP sector (71.9%).

### 2.3. Self-Report Measures

#### 2.3.1. Self-Report Measures Related to Suicidal Ideation, Planning, and Attempts

Past-year and lifetime suicidal ideation, planning, and attempts were assessed through a series of yes/no questions. Consistent with other suicide-specific research [2,3] the questions were categorised according to suicidal ideation, planning, and attempts. Suicidal ideation was assessed by asking, “Have you ever contemplated suicide?”; “Has this happened in the past 12 months?”. Suicidal planning was assessed by asking, “Have you ever made a serious plan to attempt suicide?”; “Has this happened in the past 12 months?”. Suicide attempts were assessed by asking, “Have you ever attempted suicide?”; “Did this happen in the past 12 months?”

#### 2.3.2. Self-Report Measures Related to Mental Disorders

The survey also asked participants to self-report symptoms related to various mental disorders. A ‘positive screen’ on any of the following measures indicated that the individual self-reported symptoms consistent with expectations for a diagnosis of a particular disorder. A positive screen on a self-report survey is not necessarily synonymous with actually meeting diagnostic criteria, which requires a clinical interview by a licensed professional. Individuals who used the self-report measures and indicated a positive screen should seek the evaluation of a trained and licensed clinician for the possible diagnosis of a specific mental health disorder. The current study assessed symptoms related to the mental disorders of PTSD, MDD, GAD, SAD, PD, and AUD.

PTSD symptoms were self-reported using the PTSD Checklist for DSM-5 (PCL-5) [30]. Positive screens on the PCL-5 are determined by meeting the minimum DSM-5 criteria for each PTSD cluster and exceeding the minimum clinical cut-off of greater than 32 [31,32]. The Patient Health Questionnaire (PHQ-9) was used to assess for symptoms of MDD [33,34,35]. Positive screens on the PHQ-9 were determined by a score of 9 or greater [33]. The Generalised Anxiety Disorder Scale (GAD-7) was used to assess for symptoms of GAD [36,37]. A positive screen for GAD required a total score of 9 or greater [38]. The Social Interaction Phobia Scale (SIPS) was used to assess for symptoms of SAD [39]. A positive screen for SAD requires a total score of 20 or greater [39]. The Panic Disorders Symptoms Severity Scale (PDSS-SR) was used to assess for symptoms of PD [40,41]. A positive screen for PD requires a total score of 7 or greater [40,42]. The AUDIT was used to assess for symptoms of dangerous alcohol consumption and dependence consistent with AUD [43,44,45]. A positive screen for hazardous alcohol use and dependence required the total AUDIT score to be 15 or greater [43]. Participants self-reported the measures based on the last year for the AUDIT, the past month for the PCL-5, the past 14 days for the PHQ-9 and GAD-7, and the past 7 days for the PDSS-SR. There is no specific time window used for SIPS.

### 2.4. Statistical Analyses

Several sociodemographic categories were used for comparison: (1) gender; (2) sex; (3) age; (4) education; (5) ethnicity; (6) marital status; (7) province of work; (8) previous work experience. Cross tabulations and a series of logistical regression models were used to examine the rates and association of past-year and lifetime suicidal ideation, planning, and attempts among the sociodemographic categories outlined above. A series of *t*-tests and one-way analysis of variance (ANOVA) tests were performed to compare suicidal ideation, planning, and attempts across sociodemographic categories. All tests were two-tailed with the alpha level of 0.05. Holm–Bonferroni adjustments were applied to alpha levels in post-hoc tests to decrease familywise error rate. Finally, logistic regression models were performed to evaluate the associations between suicidal behaviours (i.e., ideation, planning, attempts) and associations with mental disorders (i.e., PTSD, MDD, GAD, SAD, PD, and AUD).

## 3. Results

A considerable proportion of the total sample reported lifetime suicidal ideation (25.7%), planning (10.9%), and attempts (5.5%) in Table 1. Several participants also reported past-year suicidal ideation (7.5%), planning (2.1%), and 1 to 4 participants reported a past-year suicide attempt. No statistically significant results were observed for gender-based differences in lifetime suicidal ideation (23.3% for men; 29.3% for women) or past-year suicidal ideation (7.5% for men; 8.0% for women) in the sample.

The associated logistic regression models regarding lifetime suicidal ideation, planning, and attempts are presented in Table 2. There are relatively few statistically significant results. Individuals from 40 to 59 years in age were significantly more likely to report lifetime suicidal ideation (*p* < 0.05) than those of age 19 to 29 years old. Respondents working in Nova Scotia were statistically significantly more likely to report lifetime suicidal attempts compared to those working in British Colombia (*p* < 0.05). Individuals with previous CAF work experience and work experience in a PSP sector had statistically significantly higher reports of lifetime suicide attempts (*p* < 0.01) than those who had no such previous work experience. Other sociodemographic variable categories were not significantly associated with lifetime suicidal ideation, planning, and attempts. 

The associations between lifetime and past-year suicidal ideation, planning, attempts, and positive screens for mental disorders (i.e., PTSD, MDD, GAD, SAD, PD, AUD) are presented in Table 4. There were statistically significant associations between positive screens for PTSD, MDD, SAD and lifetime suicidal ideation, planning, and attempts. Positive screens for PD were associated with lifetime suicidal ideation and planning, but not attempts. Past-year suicidal ideation was associated with MDD (*p* < 0.01) and GAD (*p* < 0.05). Positive screens for SAD were associated with statistically significantly increased ORs in past-year suicidal planning (*p* < 0.05).

## 4. Discussion

The current study provides estimates of the percentage of lifetime and past year suicidal ideations, planning, and attempts among a sample of CCG personnel and C&P Officers. The results provide the first information extending the Canadian PSP suicide data to include CCG personnel and C&P Officers. The results indicate higher percentages of lifetime suicidal ideation, planning, and attempts, respectively, among C&P (i.e., 19.6%, 7.5%; 7.5%) and CCG (i.e., 27.5%; 12%; 4.7%) than the Canadian general population (i.e., 11.8%; 4.0%, 3.1%) [46]. Lifetime percentages of suicidal ideation, planning, and attempts among the total sample of CCG personnel and C&P Officers (i.e., 25.7%; 10.9%; 5.5%) were comparable to the overall rates reported among other Canadian PSP sectors, including nurses (i.e., 33.0%; 17.0%; 8.0%), correctional workers (i.e., 26.6%; 11.9%; 5.2%), municipal/provincial police (20.5%; 8.9%; 2.1%), RCMP (25.7%; 11.2%; 2.4%), firefighters (25.2%; 8.8%; 3.3%), paramedics (41.1%; 23.8%; 9.8%), and call centre operators/dispatchers (28.7%; 13.6%; 8.6%) [19,24,47].

The current results indicated differences in the past-year and lifetime suicidal ideation between C&P Officers (4.7%, 19.6%) and CCG personnel (8.0%, 27.5%). The differences may be due to several factors, including but not limited to the sample size, differing duties, or differences in remote deployments. Further research is required to fully understand the differences in suicidal ideation in the context of variations in occupational responsibilities and diverse sources of occupational strain.

Although not statistically significant, perhaps due to the sample size, CCG women reported higher lifetime suicidal ideation (32.2%) than women C&P Officers (17.6%). The CCG specific-sample also reported higher past-year suicidal ideation, planning, and attempts than the specific sample of C&P Officers, but the results were not statistically significant, again perhaps due to the sample size. The percentage of women who reported suicidal ideation is consistent with other PSP-specific research [19]. Wider population-based research highlights the “gender paradox” by which there is an over-representation of nonfatal suicidal ideation, planning, and attempts among women, yet fatal suicides are more common among men [48,49]. The higher percentages of suicidal behaviours reported by women highlights a specific need, and increased research may be needed to understand why men may report less suicidal ideation, planning, or attempts.

Sociodemographic covariates and lifetime suicidal ideation and attempts indicated a few statistically significant associations. Lifetime suicidal ideation was associated with age. Participants in the age categories of 40 to 49 and 50 to 59 appear to be at an elevated risk of suicidal ideation compared to participants younger than 40. The association may be partially explained by the greater likelihood of participants experiencing diverse PPTE with cumulative service and the association between frequent exposures to PPTE and mental health disorders [50]. Participants residing in Nova Scotia appear to be at an elevated risk for a suicide attempt. The possible influence of remote geography warrants additional attention and may be a proxy for several other factors including regional dynamics, population density, and access to resources. Leaders of CCG personnel and C&P Officers in Nova Scotia may consider further research and increasing focused mental health supports. Participants who reported having previous service with CAF and another PSP sector appear to be at an elevated risk for a suicide attempt. This may be a result of higher number of exposures to PPTEs and a greater likelihood to screen positive for PTSD, mood, anxiety, or acute stress disorders [51]. Participants who reported a combination of previous CAF experience and ongoing PSP service may require specifically targeted mental health support which highlights the need to inquire about previous experience during screening, assessment, and treatment in order to ensure appropriate resources are allocated [51]. All leaders may consider further research to understand mitigating variables of cumulative and/or diverse service.

The current study provides initial understandings of estimated percentages of CCG personnel and C&P Officers who reported lifetime and past-year suicidal ideation, planning, and attempts. The current results provide estimates to those supporting CCG personnel and C&P Officers. The results also provide greater nuance on suicidal ideation, planning, and attempts among other PSP populations. Nevertheless, estimating suicidal ideation, planning, and attempts remains extremely difficult as many events are not reported or do not require a hospital stay [52]. A greater understanding of the nuances surrounding suicidal ideation is required, particularly given the remote nature of the duties performed by many CCG personnel and C&P Officers. Service in profoundly remote areas is anticipated to increase amidst growing pressure for Arctic sovereignty in Canada and possible increases to the CCG budget [53,54,55]. Due to the remote nature of CCG personnel and C&P Officer service, innovative options such as PSPNET (i.e., www.pspnet.ca) (accessed on 30 October 2022), may be critical to support this PSP population.

### Strengths and Limitations

The current study has a few limitations: (1) the participation rate was approximately (18%) for C&P and (5%) for CCG, which may be due to challenges experienced at remote service locations; (2) voluntary participation created an unknowable influence from self-selection biases; (3) despite structural assurances regarding the protection of their mental health data, ongoing perceptions of the stigma of mental health challenges among PSP may have facilitated underreporting of suicidal behaviours; [8,56] (4) the self-report measure may have provided increased accuracy, but does not provide the possibility for clinicians to offer help, and if necessary, make arrangements to assist those at an elevated risk of a suicide attempt; (5) data were collected during COVID-19 which may have compounded additional stressors related to past-year suicidal ideation, attempts, and planning; however, the extent to which COVID played a role remains uncertain.

The study limitations are offset by several strengths: (1) web-based surveys, which allowed even CCG personnel and C&P Officers operating in remote locations the possibility to participate; (2) the ability of respondents to self-report suicidal ideation, planning, and attempts without the presence of clinicians, which may provide more accurate data due to possible stigma; (3) the respondents were asked both past-year and lifetime suicidal ideation, planning, and attempts.

## 5. Conclusions

The current study provides a first estimate of the percentage of suicidal ideation, planning, and attempts among C&P Officers and CCG within the past year and lifetime. The percentages appear higher than the Canadian general population, but comparable to other Canadian PSP communities. There were statistically significant associations between past-year suicidal ideation and positive screens for MDD and GAD, and past-year suicidal planning showed statistically significant associations with SAD. Lifetime suicidal ideation, planning, and attempts were statistically significantly associated with positive screens for PTSD, MDD, and SAD. Additional research and efforts are needed to support specific members within the CCG personnel and C&P Officers who may be at greater risk (i.e., individuals older than 40 years, women). Further exploration and mental health support planning may be required to support CCG personnel and C&P Officers in Nova Scotia due to higher reported attempts than other provinces. Continued support for former CAF veterans is needed, particularly for individuals with career experience in a different PSP sector.

## Figures and Tables

**Table 1 ijerph-19-16318-t001:** Percentage of Lifetime and Past-Year Suicidal ideation, planning, and attempts (*n* = 385).

		Lifetime			Past-Year		
		Ideation	Planning	Attempts	Ideation	Planning	Attempts
		% (*n*)	% (*n*)	% (*n*)	% (*n*)	% (*n*)	% (*n*)
Total Sample	All (*n* = 385)	25.7 (99)	10.9 (42)	5.5 (21)	7.5 (29)	2.1 (8)	^
	CCG (*n* = 276)	27.5 (76)	12.0 (33)	4.7 (13)	8.0 (22)	2.5 (7)	^
	C&P (*n* = 107)	19.6 (21)	7.5 (8)	7.5 (8)	4.7 (5)	^	-
Men	All (*n* = 227)	23.3 (53)	9.7 (22)	4.8 (11)	7.5 (17)	^	^
	CCG (*n* = 155)	23.9 (37)	11.6 (18)	3.2 (5)	8.4 (13)	^	^
	C&P (*n* = 71)	21.1 (15)	^	8.5 (6)	^	-	-
Women	All (*n* = 150)	29.3 (44)	13.3 (20)	6.7 (10)	8.0 (12)	^	-
	CCG (*n* = 115)	32.2 (37)	13.0 (15)	7.0 (8)	7.8 (9)	^	-
	C&P (*n* = 34)	17.6 (6)	^	^	^	^	-

Notes: ^ = The sample size is between 1 and 4; consequently, the data cannot be presented; - = no data reported; All = total sample; CCG = Canadian Coast Guard; C&P = Conservation and Protection.

**Table 2 ijerph-19-16318-t002:** Associations between Sociodemographic Covariates and Lifetime Suicidal ideation, planning, and attempts.

	Total (*n* = 385)	Ideation		Planning		Attempts	
	% (*n*)	% (*n*)	OR (95% CI)	% (*n*)	OR (95% CI)	% (*n*)	OR (95% CI)
Gender							
Men	59.0 (227)	23.3 (53)	1	9.7 (22)	1	4.8 (11)	1
Non-binary	1.3 (5)	^	^	-	-	-	-
Two-spirit	-	-	-	-	-	-	-
Women	39.0 (150)	29.3 (44)	1.36 (0.85, 2.17)	13.3 (20)	1.45 (0.76, 2.75)	6.7 (10)	1.40 (0.58, 3.37)
Sex							
Male	59.7 (230)	23.9 (55)	1	10.0 (23)	1	4.3 (10)	1
Female	39.7 (153)	28.8 (44)	1.28 (0.81, 2.04)	12.4 (19)	1.29 (0.67, 2.45)	7.2 (11)	1.70 (0.70, 4.10)
Age							
19–29	12.7 (49)	14.3 (7)	1	^	-	^	-
30–39	28.8 (111)	25.2 (28)	2.02 (0.82, 5.02)	10.8 (12)	-	6.3 (7)	-
40–49	28.1 (108)	30.6 (33)	2.64 (1.08, 6.49) *	16.7 (18)	-	6.5 (7)	-
50–59	23.6 (91)	29.7 (27)	2.53 (1.01, 6.34) *	8.8 (8)	-	5.5 (5)	-
60+	5.5 (21)	^	1.41 (0.34, 5.45)	^	-	^	-
Education							
High School or Less	9.1 (35)	17.1 (6)	1	^	-	^	-
College Program (e.g., Trade School; 2-Year College Diploma)	39.7 (153)	28.1 (43)	1.89 (0.73,4.87)	14.4 (22)	-	7.8 (12)	-
Coast Guard College: Graduated Fleet	10.1 (39)	23.1 (9)	1.45 (0.46,4.59)	^	-	^	-
Coast Guard College: MCTS Officer Training	2.3 (9)	^	2.42 (0.47,12.47)	^	-	^	-
University Degree (4-year College or Higher)	33.5 (129)	27.1 (35)	1.80 (0.69,4.70)	7.8 (10)	-	5.4 (7)	-
Ethnicity							
Asian	2.3 (9)	^	-	^	-	^	-
Black	^	^	-	^	-	^	-
Hispanic	^	^	-	^	-	^	-
Indigenous (i.e., First Nations, Inuit, Métis)	3.4 (13)	^	-	^	-	^	-
South Asian	^	^	-	^	-	^	-
Caucasian	88.1 (339)	25.1 (85)	-	10.0 (34)	-	5.3 (18)	-
Prefer Not to Answer	1.3 (5)	^	-	^	-	^	-
Other	3.9 (15)	53.3 (8)	-	33.3 (5)	-	^	-
Marital Status							
Single	22.1 (85)	27.1 (23)	1	10.6 (9)	1	^	-
Married/Common Law	68.1 (262)	23.3 (61)	0.82 (0.47, 1.43)	9.9 (26)	0.93 (0.42, 2.07)	5.7 (15)	-
Separated/Divorced	7.5 (29)	44.8 (13)	2.19 (0.91, 5.25)	17.2 (5)	1.76 (0.54, 5.76)	^	-
Widowed	^	^	2.70 (0.16, 44.90)	^	8.44 (0.49, 146.96)	^	-
Province of Work							
British Columbia	56.4 (217)	76.0 (165)	1	10.6 (23)	1	4.1 (9)	1
New Brunswick	1.8 (7)	^	0.53 (0.06, 4.49)	^	-	^	-
Newfoundland and Labrador	7.3 (28)	25.0 (7)	1.06 (0.43, 2.63)	^	1.01 (0.28, 3.62)	^	-
Northern Territories (YK, NWT, NVT)	^	^	-	^	-	^	-
Nova Scotia	9.6 (37)	35.1 (13)	1.72 (0.82, 3.62)	13.5 (5)	1.32 (0.47, 3.72)	13.5 (5)	3.59 (1.13, 11.41) *
Ontario	11.7 (45)	37.8 (17)	1.93 (0.98, 3.80)	15.6 (7)	1.60 (0.64, 3.99)	^	1.64 (0.43, 6.33)
Québec	12.7 (49)	18.4 (9)	0.71 (0.32, 1.57)	^	0.75 (0.25, 2.28)	^	2.04 (0.60, 6.93)
Previous Work Experience							
Neither	71.9 (277)	23.8 (66)	1	10.1 (28)	1	4.3 (12)	1
Public Safety Only	16.9 (65)	30.8 (20)	1.42 (0.78, 2.58)	12.3 (8)	1.24 (0.54, 2.87)	7.7 (5)	1.83 (0.62, 5.40)
CAF Only	8.6 (33)	24.2 (8)	1.02 (0.44, 2.38)	^	0.57 (0.13, 2.52)	^	0.69 (0.09, 5.46)
CAF and Public Safety	2.6 (10)	50.5 (5)	3.20 (0.90, 11.38)	^	5.91 (1.57, 22.20)	^	9.43 (2.17, 41.05) **
Occupation Category							
CCG	71.7 (276)	27.5 (76)	1	12.0 (33)	1	4.7 (13)	1
C&P	27.8 (107)	19.6 (21)	0.64 (0.37, 1.11)	7.5 (8)	0.59 (0.26, 1.33)	7.5 (8)	1.63 (0.66, 4.05)

Notes: ^ = Sample size is between 1 and 4; consequently, the data cannot be presented; - = no data reported; * *p* < 0.05, ** *p* < 0.01—Statistically significantly different; CAF = Canadian Armed Forces; CCG = Canadian Coast Guard; C&P = Conservation and Protection; MCTS = Marine Communications and Traffic Services. The associated logistic regression models regarding past-year suicidal ideation, planning, and attempts are presented in Table 3.

**Table 3 ijerph-19-16318-t003:** Associations between Sociodemographic Covariates and Past-Year Suicidal ideation, planning, and attempts.

	Ideation		Planning		Attempts	
	% (*n*)	OR (95% CI)	% (*n*)	OR (95% CI)	% (*n*)	OR (95% CI)
Gender						
Men	7.5 (17)	1	^	-	^	-
Non-binary	^	^	-	-	-	-
Two-spirit	-	-	-	-		
Women	8.0 (12)	0.79 (0.33, 1.91)	^	-	^	-
Sex						
Male	7.8 (18)	1	^	-	^	-
Female	7.2 (11)	0.69 (0.28, 1.66)	^	-	^	-
Age						
19–29	^	-	^	-	^	-
30–39	9.0 (10)	-	^	-	^	-
40–49	6.5 (7)	-	^	-	^	-
50–59	11.0 (10)	-	^	-	^	-
60 +	^	-	^	-	^	-
Education						
High School or Less	^	-	^	-	-	-
College Program (e.g., Trade School; 2-Year College Diploma)	9.8 (15)	-	^	-	-	-
Coast Guard College: Graduated Fleet	^	-	^	-	-	-
Coast Guard College: MCTS Officer Training	^	-	^	-	-	-
University Degree (4-year College or Higher)	7.8 (10)	-	^	-	-	-
Ethnicity						
Asian	^	-	^	-	^	-
Black	^	-	^	-	^	-
Hispanic	^	-	^	-	^	-
Indigenous (i.e., First Nations, Inuit, Métis)	^	-	^	-	^	-
South Asian	^	-	^	-	^	-
Caucasian	7.4 (25)	-	2.1 (7)	-	^	-
Prefer Not to Answer	^	-	^	-	^	-
Other	^	-	^	-	^	-
Marital Status						
Single	9.4 (8)	1	^	-	-	-
Married/Common Law	5.0 (13)	0.51 (0.18, 1.46)	^	-	-	-
Separated/Divorced	20.7 (6)	1.61 (0.40, 6.44)	^	-	-	-
Widowed	^	-	^	-	-	-
Province of Work						
British Columbia	8.3 (18)	1	2.8 (6)	1	^	-
New Brunswick	^	-	^	-	^	-
Newfoundland and Labrador	^	0.76 (0.13, 4.29)	^	-	^	-
Northern Territories (YK, NWT, NVT)	^	-	^	-	^	-
Nova Scotia	^	0.57 (0.14, 2.32)	^	-	^	-
Ontario	^	0.58 (0.17, 2.05)	^	0.47 (0.05, 4.77)	^	-
Québec	^	0.54 (0.10, 2.87)	^	0.94 (0.08, 10.91)	^	-
Previous Work Experience						
Neither	7.2 (20)	1	1.8 (5)	1	^	-
Public Safety Only	10.8 (7)	1.24 (0.43, 3.57)	^	1.53 (0.24, 9.95)	^	-
CAF Only	^	0.33 (0.04, 2.85)	^	-	^	-
CAF and Public Safety	^	0.58 (0.06, 5.47)	^	1.53 (0.13, 17.97)	^	-
Occupation Category						
CCG	8.0 (22)	1	2.5 (7)	1	^	-
C&P	4.7 (5)	0.77 (0.25, 2.35)	^	0.53 (0.06, 5.06)	^	-

Notes: ^ = Sample size is between 1 and 4; consequently, the data cannot be presented; - = no data reported; CAF = Canadian Armed Forces; CCG = Canadian Coast Guard; C&P = Conservation and Protection; MCTS = Marine Communications and Traffic Services. No statistically significant results were shown for the sociodemographic covariates and reported past-year suicidal ideation, planning, or attempts.

**Table 4 ijerph-19-16318-t004:** Associations Between Past-Year and Lifetime Suicidal ideation, planning, and attempts and Mental Disorders.

	PTSD	MDD	GAD	SAD	PD	AUD
	OR (95% CI)	OR (95% CI)	OR (95% CI)	OR (95% CI)	OR (95% CI)	OR (95% CI)
Lifetime						
Ideation	5.14 (2.87, 9.19) ***	3.13 (1.90, 5.16) ***	1.74 (0.96, 3.15)	2.63 (1.55, 4.46) ***	2.15 (1.02, 4.55) *	1.88 (0.82, 4.33)
Planning	4.94 (2.46, 9.93) ***	2.68 (1.38, 5.19) **	1.62 (0.73, 3.61)	3.56 (1.80, 7.05) ***	2.65 (1.06, 6.64) *	1.47 (0.48, 4.53)
Attempt	4.15 (1.62, 10.65) **	3.72 (1.52, 9.06) **	1.72 (0.61, 4.90)	3.64 (1.48, 8.91) **	2.56 (0.81, 8.14)	1.31 (0.29, 5.96)
Past-Year						
Ideation	2.12 (0.86, 5.26)	5.19 (2.03, 13.25) ***	3.54 (1.30, 9.70) *	1.64 (0.66, 4.08)	2.33 (0.70, 7.72)	0.53 (0.11, 2.69)
Planning	3.06 (0.62, 15.06)	5.50 (0.96, 31.59)	2.70 (0.50, 14.53)	10.50 (1.13, 97.91) *	4.05 (0.69, 23.90)	-
Attempt	-	-	-	-	-	-

Note: OR- Odds ratios; CI- confidence interval; * *p* < 0.05, ** *p* < 0.01, *** *p* < 0.001—Statistically significantly different from the reference group; - = no data reported; AUDIT = Alcohol Use Disorder; CCG = Canadian Coast Guard; C&P = Conservation and Protection; GAD = Generalized Anxiety Disorder; MDD = Major Depressive Disorder; PTSD = Posttraumatic Stress Disorder; PD = Panic Disorder; SAD = Social Anxiety Disorder.

## Data Availability

Data access will not be provided due to the sensitive nature of the content.

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
