# Peer review of "Suicidal Ideation, Planning, and Attempts among Canadian Coast Guard and Conservation and Protection Officers"

_ijerph, 2022, doi:10.3390/ijerph192316318_

Round 1

Reviewer 1 Report

Questions and suggestions:

Question 1: Line 123 "Participants who completed at least 30% of the survey were included in the current analyses and resultsWhy 30%? which is the criteria to chose this percentage? 

Suggestion 1: The methodology is well explained and the Ethics section includes the code of Research Ethics Board bur there are not information about informed consent. This should be attached or at least explained. 

Suggestion 2: The description of the statistical analyses should include how variables are expressed, for example: frequency and percentage.

Suggestion 3: The conclusion is very weak. It should be much more consistent, since both in results and discussion there is enough information to support a much clearer and more concrete conclusion. I suggest modifying the wording

Author Response

December 1, 2022

Re: Manuscript Revisions 

Title: Suicidal Ideation, Planning, and Attempts among Canadian Coast Guards and Conservation & Protection Officers

Authors: Nisbet, J., Jamshidi, L., Andrews, K.L., Teckchandani, T.A., Price, J.A.B., Ricciardelli, R., Anderson, G.S., and Carleton, R. N.

Dear Ms. Xiong and the reviewers,

We hope this letter finds each of you well.

Thank you to the reviewers for their comments and for the opportunity to revise our manuscript. We have endeavoured to address the reviewers' feedback in detail below. 

Reviewer 1

Question 1: Line 123 "Participants who completed at least 30% of the survey were included in the current analyses and results" Why 30%? which is the criteria to chose this percentage

Thank you for your question, Reviewer 1. Thirty percent is the standard cut-off used in the previous Public Safety Personnel research and allows for direct comparison with the previously collected data (i.e., Carleton et al., 2018).

Please note the manuscript has been updated:

"Participants who completed at least 30% or more of the survey were included in the current analyses and results."

Suggestion 1: The methodology is well explained and the Ethics section includes the code of Research Ethics Board bur there are not information about informed consent. This should be attached or at least explained. 

We appreciate your suggestion. The manuscript has been updated:

"All participants were provided with a plain language summary of the research and completed an informed consent."

Informed consent is also stated at the end of the manuscript.

Suggestion 2: The description of the statistical analyses should include how variables are expressed, for example: frequency and percentage.

Thank you for your comment. Please note the %(n) are presented at the top of the tables.

Suggestion 3: The conclusion is very weak. It should be much more consistent, since both in results and discussion there is enough information to support a much clearer and more concrete conclusion. I suggest modifying the wording

Thank you for your suggestion. The conclusion has been updated:

The current study provides a first estimate of the percentage of suicidal ideation, planning, and attempts among C&P Officers and CCG within the past-year and lifetime. The percentages appear higher than the Canadian general population, but comparable to other Canadian PSP communities. There were statistically significant associations between past-year suicidal ideation and positive screens for MDD and GAD, and past-year suicidal planning showed statistically significant associations with positive screens for SAD. Lifetime suicidal ideation, planning, and attempts were statistically significantly associated with positive screens for PTSD, MDD, and SAD. Additional research and efforts are needed to support specific members within the CCG personnel and C&P Officers who may be at greater risk (i.e., individuals older than 40, women). Further exploration and mental health support planning may be required to support CCG personnel and C&P Officers in Nova Scotia due to higher reported attempts than in other provinces. Continued support for former CAF veterans is needed, particularly individuals with career experience in a different PSP sector.

Reviewer 2

Dear authors, I am sure that a little clarification should be made in the manuscript about why you are studying this sample (Coast Guards and Conservation & Protection Officers). In my opinion, the Introduction does not sufficiently clarify why suicidal tendencies should be studied in this sample. This clarification will allow other researchers to compare the results of your research with their own on a larger scale. In the current format, the article looks interesting mostly for local researchers, and this may become a limitation. The same is about Discussion: I didn't understand why your results are more important for these professional groups than for others. Otherwise, I rate the quality of the article very highly.

Thank you for your helpful comments and high rating.

Please note we have further clarified the language in the introduction. The manuscript now states:

Despite indications of increased risk for suicidal ideation, planning, and attempts as in other PSP sectors, the literature focused on CCG personnel and C&P Officers is limited, with no published results regarding suicidal behaviours [19]. Comparisons may be possible with coast guard personnel from the United States Coast Guard (USCG) due to similar occupational duties; however, there may be important differences because the USCG is overseen by the Department of Defence whereas in Canada CCG personnel operate as a special agency within Fisheries and Oceans Canada [14]. In 2018, the USCG estimated the past-year suicidal ideation (4.7%), planning (1.6%), and attempts (0.2%) [21]. Previous research emphasised high levels of secrecy within the USCG and ongoing mental health stigma may prevent personnel from seeking help, alongside adverse interpersonal or professional costs [22].

The article may be of interest to other national or international PSP organisations. As you indicate, it is a baseline study for other organisations. The study of suicidal ideation, planning, and attempts remains limited, and we believe this manuscript contributes to the broader literature on public safety personnel.

We want to extend our thanks to the reviewers for their helpful comments. We hope that we have adequately addressed the concerns. We hope the revisions have resulted in a manuscript suitable for publication.

We look forward to your editorial feedback and decision. If you have any questions, please contact us by phone (1-306-540-2064) or e-mail (Jolan.Nisbet@uregina.ca). 

With kind regards,

Jolan Nisbet

Reviewer 2 Report

Dear authors, I am sure that a little clarification should be made in the manuscript about why you are studying this sample (Coast Guards and Conservation & Protection Officers). In my opinion, the Introduction does not sufficiently clarify why suicidal tendencies should be studied in this sample. This clarification will allow other researchers to compare the results of your research with their own on a larger scale. In the current format, the article looks interesting mostly for local researchers, and this may become a limitation. The same is about Discussion: I didn't understand why your results are more important for these professional groups than for others.  Otherwise, I rate the quality of the article very highly.

Author Response

(The authors gave the same response as above.)
